# What Is the Real Influence of Climatic and Environmental Factors in the Outbreaks of African Swine Fever?

**DOI:** 10.3390/ani12060781

**Published:** 2022-03-19

**Authors:** Andrei Ungur, Cristina Daniela Cazan, Luciana-Cătălina Panait, Mircea Coroian, Cornel Cătoi

**Affiliations:** 1Department of Pathology, Faculty of Veterinary Medicine, University of Agricultural Sciences and Veterinary Medicine of Cluj-Napoca, 400372 Cluj-Napoca, Romania; cornel.catoi@usamvcluj.ro; 2Molecular Biology and Veterinary Parasitology Unit (CDS-9), Faculty of Veterinary Medicine, University of Agricultural Sciences and Veterinary Medicine of Cluj-Napoca, 400372 Cluj-Napoca, Romania; cristina.cazan@usamvcluj.ro; 3Department of Parasitology and Parasitic Diseases, Faculty of Veterinary Medicine, University of Agricultural Sciences and Veterinary Medicine of Cluj-Napoca, 400372 Cluj-Napoca, Romania; luciana.rus@usamvcluj.ro (L.-C.P.); mircea.coroian@usamvcluj.ro (M.C.); 4Technological Transfer Centre for Animal Nutrition and Comparative Pathology ‘COMPAC’, University of Agricultural Sciences and Veterinary Medicine of Cluj-Napoca, 400372 Cluj-Napoca, Romania

**Keywords:** ASF, geoclimatic, transmission, Romania, statistical analysis

## Abstract

**Simple Summary:**

African Swine fever is present on the African, European and Asian continents, causing devastating economic impact due to the mandatory mass depopulation in confirmed outbreaks. Using the data reported by the Romanian National Sanitary Veterinary and Food Safety Authority, we created a database with all the localities from Romania where the outbreaks were confirmed between 2020 and 2021 as well as the geoclimatic features of those areas. The database was then used to statistically analyze the frequency of confirmed outbreaks in relation with specific climatic and environmental factors. Our results show that such studies need to be continued in order to have a better understanding of the geoclimatic risk factors in the transmission of African swine fever virus.

**Abstract:**

African swine fever has a significant economic, social and environmental impact due to official regulation of the disease, namely the mass depopulation of all pigs in confirmed outbreaks. The main objective of the present study was to statistically analyze the possible correlation between the number of outbreaks and infected pigs from backyard farms with the altitude, seasonal average annual humidity, and average temperature during spring, summer, autumn and winter, as well as the distance from forests, rivers, and lakes in Romania. The study included all infected backyard pigs with African swine fever virus (*n* = 7764) and outbreaks (*n* = 404) that occurred in Romania between 6 February 2020 and 2 March 2021. The number of infected pigs and ASF outbreaks were significantly higher in localities at altitudes between 100 and 500 m, average annual humidity between 60% and 80%, average spring temperature between 10 and 14 °C, average summer temperature between 16 and 22 °C, average autumn temperature between 10 and 13 °C, average winter temperatures between −1 and 2 °C, and at distances of less than 5 km from the forests, less than 15 km from lakes and less than 5 km from the rivers. The number of affected pigs decreased significantly at summer temperatures below 16 °C.

## 1. Introduction

African swine fever (ASF) is a viral hemorrhagic pig illness that has spread throughout Europe, wreaking havoc on pig production and the economy, disrupting trade in pigs and porcine products, and even affecting social welfare in affected areas [1]. African swine fever virus (ASFV), the disease’s causal agent, is a big, complex, enveloped DNA virus that belongs to the *Asfarviridae* family and the *Asfivirus* genus and is known for its exceptional environmental stability [2,3]. Infectious ASFV can be recovered from pig tissues for months or even years at low temperatures, notably from blood, muscle, and skin tissues stored at −20 or 4 °C [4]. The disease affects both wild and domestic pigs, most often with acute evolution. The infected animals present hyperthermia, anorexia, skin hyperemia, respiratory and vascular disorders. The mortality rate can reach 100% [5]. Montgomery first reported the disease in Kenya in 1921 [6], but the disease is now widespread in Africa, Europe, and Asia, causing massive economical and zootechnical losses [7]. In 2017, the disease was also reported in Romania, the first case being reported in two domestic pigs from Satu Mare County, in the northwest of the country. A year later, it was reported in the Danube Delta, also in domestic pigs, and then it was confirmed in wild boars. The emergence of the virus was attributed to the illegal trade of meat and pork products in Satu Mare, and due to the migration of wild boars near country borders in the Danube Delta [8]. Since Romania has high pork meat consumption, pig farming is an important branch of agriculture in the country, and the economic losses are being felt by both backyard and large industrial farms. In order to provide warnings or advice to regulatory agencies responsible for the protection of public health in general and animal health, especially in terms of trends in the transmission of diseases in herds, veterinary epidemiology helps by studying the spread of diseases in livestock [9]. External biosecurity risk factors have been thought to have a substantial influence in the introduction and spread of ASF in domestic pig herds. Biosecurity measures are thought to be a fundamental instrument for preventing the introduction and spread of illnesses in animal populations, and they have been demonstrated to be so. Many risk factors connected to biosecurity have been found to have a crucial role in the spread and maintenance of ASF in domestic pig populations [10].

The aim of the study was to highlight the epidemiological aspects encountered among the domestic swine population affected by the ASFV in Romania and their possible association with new outbreaks of the disease. The study describes the correlation between the number of outbreaks and the number of affected pigs with the climatic and environmental characteristics of the outbreak areas, such as average annual humidity, average seasonal temperature, altitude, and distance from forests, lakes, and rivers. However, these observations could form a basis for a raising hypothesis about the potential climatical and environmental risk factors for ASF introduction into pig herds that could be further investigated.

## 2. Materials and Methods

The study was conducted taking into account all ASF outbreaks that occurred in Romania between 6 February 2020 and 2 March 2021, and it was confirmed by the National Sanitary Veterinary and Food Safety Authority (ANSVSA) [11]. A total of 409 outbreaks from 295 different localities were registered. The total number of confirmed infected domestic pigs was 94197, originating from both backyard (*n* = 7764) and industrial (*n* = 86,433) farms. The highest number of outbreaks was reported in pigs raised in backyard (98.7%, CI: 97.71–99.84, *n* = 404), compared with those from industrial farms (1.3%, CI: 0.15–2.29, *n* = 5).

The following epidemiological data were taken into account for each outbreak locality: average relative humidity (%), average temperature during spring, summer, autumn and winter (°C), altitude (m), distance from the closest forest (km), distance from the closest lake (km), distance from the closest river (km), and geographical location (county, latitude and longitude). Month and season of disease confirmation were also considered. The measurement of the distance between the nearest locality boundary and forest, river, or lake was made based on the mapping provided by the Google Maps online service. Data on average annual humidity were obtained from the CliMond database [12], while the data on the average annual temperature during spring, summer, autumn and winter and altitude were obtained from a free digital elevation model provided by the WorldClim website [13].

The Shapiro–Wilk normality test was used to evaluate data distribution. The statistical analysis was performed taking into account the number of pigs affected by ASFV and the number of outbreaks in each locality, respectively. For the statistical evaluation of the factors that influenced the occurrence of new cases or outbreaks of ASF, data regarding industrial farms were not included in the analyses. The large number of cases, with an increased homogeneity of their epidemiological characteristics, would have negatively influenced data distribution and, consequently, the results of the statistical analysis. Therefore, the impact of altitude, origin (county and geographical coordinates: latitude and longitude), month and season of positivity confirmation, environmental factors (average annual humidity and average temperature during spring, summer, autumn and winter), and distances from the closest forest, river, or lake on the number of affected pigs and identified outbreaks were assessed using the Spearman correlation. The correlation strength was evaluated based on the Spearman correlation coefficient (*r_s_*) (0.00–0.19: very weak; 0.20–0.39: weak; 0.40–0.59: moderate; 0.60–0.79: strong; 0.80–1.0: very strong). Using the same data, linear regression analyses were performed to establish the existence of a linear ratio. These relationships were interpreted based on the regression coefficient R^2^. Kruskal–Wallis rank sum test and Chi-square goodness of fit were used to compare the number of affected individuals and the number of outbreaks with categorical variables. A *p*-value less than 0.05 was considered statistically significant. Data were analyzed using R v. 4.0.5 software.

## 3. Results

A database was created containing a total of 295 localities in which outbreaks of ASF have been confirmed in domestic pigs, from 35 counties (Appendix A). Thus, a total number of 94,197 affected pigs was identified, of which 86,433 came from industrial farms (*n* = 5), and 7764 were raised extensively or in authorized natural farms (*n* = 404) (Figure 1). For the statistical analysis of the associations between the number of affected pigs and the climatical and environmental factors, only the 7764 affected pigs were taken into account. The highest number of affected pigs was identified in Arad County, (*n* = 1867), followed by Gorj County (*n* = 900) and Bihor County (*n* = 630). The lower number of affected pigs was identified in Galați County (*n* = 2), Bacău County (*n* = 3), Caraș-Severin County (*n* = 4) and Ialomița County (*n* = 4), while no cases of infection in backyard farms or authorized natural farms were identified in Brașov, Brăila, Călărași, Harghita, Hunedoara, and Tulcea counties. Following the statistical analysis, the number of affected pigs in Arad County was significantly higher than in the rest of the counties, at a *p*-value lower than 0.001. A total of 409 outbreaks were identified in 36 counties in Romania. In Gorj county, a significantly higher number (*n* = 66) of outbreaks was identified compared to the rest of the counties, at a *p*-value of 0.027. In Arad County, 43 outbreaks were observed, being the second county with the most outbreaks and the first with the largest number of affected pigs in Romania (Table 1).

Statistically significant differences were identified between the number of cases and outbreaks in each county, at a *p*-value of 0.007 and 0.008, respectively (Kruskal–Wallis rank sum test, Kruskal–Wallis chi-squared = 58.491 for the number of cases, and Kruskal–Wallis chi-squared = 58.09 for the number of outbreaks). A weak positive correlation was detected between the number of cases and the number of outbreaks (Spearman correlation, *r_s_* = −0.35, *p* < 0.001).

A very weak negative correlation was identified between longitude and the number of cases (Spearman correlation, *r_s_* = −0.19, *p* = 0.001) or the number of outbreaks (Spearman correlation, *r_s_* = −0.16, *p* = 0.006), with more cases of infection in the western part of Romania. A positive linear regression was detected between the number of cases and longitude (*p* = 0.02, R^2^= 0.013), showing an increase in the number of cases in the west of the country. A very weak negative correlation (Spearman correlation, *r_s_* = −0.13, *p* = 0.01) was identified between the number of affected pigs and longitude.

A significantly lower number of affected pigs was noted in April compared to the other months (Chi-square goodness of fit, χ^2^ = 9630.4, *p* < 0.001). A significantly lower number of outbreaks was observed in April and May (Chi-square goodness of fit, χ^2^ = 433.67, *p* < 0.001) (Figure 2C,D).

The association between the number of ASFV-positive pigs or the number of ASF outbreaks with the average annual humidity showed that the vast majority of cases or outbreaks was reported in localities with humidity levels between 60% and 80%, at a value *p* < 0.001. On the contrary, the lowest number of affected pigs was observed at a humidity between 50–60% (Figure 3A).

Following the statistical analysis, it was found that both the number of affected pigs and the number of outbreaks were found to be higher in localities with an average spring temperature of 10–14 °C. The number of infected pigs is much lower (*p* < 0.001) in the localities with an average spring temperature between 6 and 10 °C (Figure 3B). During spring, a significantly lower number of affected pigs was recorded when compared to the other seasons (Chi-square goodness of fit, χ^2^ = 9403.8, *p* < 0.001) (Figure 2B–D). A positive linear regression was identified between the number of cases and the average temperature during spring, at a *p*-value of 0.049. However, a regression coefficient of only 0.009 was detected showing a weak predictability. A significantly lower number of affected pigs was noted in April compared to the other months (Chi-square goodness of fit, χ^2^ = 9630.4, *p* < 0.001). A significantly lower number of outbreaks was observed in April and May (Chi-square goodness of fit, χ^2^ = 433.67, *p* < 0.001) (Figure 2C,D). Moreover, summer temperatures between 16 and 22 °C were associated with the highest number of ASF cases and outbreaks, this number decreasing significantly (*p* < 0.001) at summer temperatures below 16 °C (Figure 3C). Statistical analysis showed that both the number of affected pigs and the number of outbreaks were higher when the average autumn temperatures ranged between 10 and 13 °C, when compared to temperatures of 7–10 °C (*p* < 0.001). In winter, a higher number of outbreaks and cases of infection was identified at temperatures between −1 and 2 °C, than at temperatures below −1 °C (*p* < 0.001). A higher number of outbreaks was noticed during winter, at a *p* value lower than 0.001 (Chi-square goodness of fit, χ^2^ = 395.55) (Figure 2K–M).

It was observed that the incidence of ASF-positive individuals and outbreaks was significantly higher in the localities at altitudes between 100 and 500 m. The localities situated at lower altitudes (0–100 m) and higher altitudes (500–1400 m) had a lower number of cases and outbreaks, at a *p*-value lower than 0.001. The highest number of outbreaks encountered in a single locality (*n* = 9) was in Roșia de Amaradia locality from Gorj County, located at an altitude of 452 m (Appendix A).

After the comparison between the number of affected pigs and the distance from the forest, it was observed that the largest share of affected pigs was at a distance lower than 5 km from the forest. At distances greater than 20 km, only 95 affected pigs were identified out of a total of 7534 positive pigs, at a significant value of (*p* < 0.001). The largest number of outbreaks was also identified at distances lower than 5 km from the forest (*p* < 0.001). At distances greater than 10 km, only 16 outbreaks were identified (Figure 4A). 

A significantly higher number of affected pigs was identified at distances from lakes of lower than 15 km, at a value of *p* < 0.001. At distances greater than 30 km, only 235 affected pigs were identified (Figure 4B). The same situation was observed in the case of outbreaks (*p* < 0.001). The number of outbreaks was higher near lakes (linear regression, *p* = 0.003, R^2^ = 0.03). Similarly, the highest number of outbreaks and affected pigs was observed at a distance of less than 5 km from the river (*p* < 0.001) (Figure 4C).

Spearman correlations were used to analyze the risk factors, which assessed the presence of correlations between geographical factors (average annual humidity, average temperature during spring, summer, autumn and winter, altitude, distance from forest, lake, and river) and the number of affected pigs and the number of outbreaks in each locality. Linear regression analysis was used to establish a linear relationship between climatic factors (average annual humidity, temperature during spring and summer), environmental factors (altitude, the distance between forest, lake, and river) and the number of affected pigs, or the number of outbreaks in each locality. A positive linear regression was identified between the number of outbreaks and the distance from the nearest lake at a *p* value of 0.004 (Figure 4D). However, the correlation coefficient R^2^ (0.024) was low, indicating a low-intensity predictability relationship. For the other values analyzed, no linear mathematical relationships of predictability were identified. Moreover, no statistically significant results were obtained when Kruskal–Wallis ranks sum test was used.

## 4. Discussion

ASF can be considered the most important disease of pigs during our times. It was reported for the first time in Romania 4 years ago [8]. In the present study, the spreading direction of ASFV was not clear, as many cases were already reported in all macro regions of Romania [11]. The current data were assessed after three years since the first ASFV confirmation in the country (Figure 2). In addition, in Romania, most of the ASFV infections occurred in backyard farms and rarely in the industrial ones, possibly correlated with the higher number of backyard farms compared to industrial farms. All types of farming systems have their particular set of risk factors. Industrial farms usually have the best biosecurity practices in place, but they might experience unintended biosecurity breaches. Conversely, backyard farms usually experience poor biosecurity levels. Possible explanations could be the lack of ASF awareness for the breeders, other sociocultural factors such as traditional pork consumption and possible wild boar contact with domestic pigs [5].

In the absence of a commercially available vaccine, it is important to limit the spread of the virus as much as possible by interrupting its routes of transmission. In Eastern Europe, outbreaks in pig farms have been reported mainly in areas where the virus circulates in the wild boar population [8,14].

In 2014, Chenais demonstrated direct transmission between wild boars, as well as indirect transmission between wild boars via habitat, suggesting that the wild boar–habitat cycle dominated the spread of ASF virus in Eastern Europe [15]. Later, outbreaks in domestic pigs began to be correlated with cases of wild boar infection [14]. This could be explained by the high concentrations of ASFV in meadows and crop fields due to excretions, blood, and carcasses of wild boars [16]. However, in the summer of 2018, in Romania, many more outbreaks were diagnosed among domestic pigs (*n* = 1073), compared to the number of cases reported in wild boars (*n* = 155) [8]. This could be explained by the lower chance to find and observe all wild boars in their natural habitat.

In this study, climatical and environmental factors that could favor the occurrence, spread, and distribution of ASFV in domestic pigs were considered. Therefore, the correspondence between the geographical factors and the number of affected domestic pigs and the number of outbreaks of ASF in Romania, from February 2020 to March 2021, were analyzed. Based on the Spearman correlation test, the correlations between the number of affected pigs in each locality and the number of outbreaks in the same locality were observed. Following the statistical analysis, a linear regression was noticed between the number of outbreaks and the distance from the lake, suggesting a predictive relationship between the two variables. The same results were obtained in a previous study [17] in Romania, in which the risk factors that could influence the occurrence of ASF cases in domestic pigs and wild boars were analyzed, in the period from 2018–2019. Thus, the existence of positive correlation between the density of rivers and forests and the presence of infection in the pig population studied, both for 2018 and 2019, was noted [17]. Another study, demonstrated in Russia that aquatic environments could be a risk factor in the occurrence of ASF outbreaks in 2007–2010 [18]. In the present study, a positive correlation was also found between the number of ASFV-infected pigs/ASF outbreaks and a lower distance from lakes (<15 km) or rivers (<5 km). An epidemiological study conducted by EFSA, in 2017, which evaluated the risk factors associated with the infection in the wild boar population, showed a positive correlation between the proportion of lakes and the occurrence of ASF in Latvia (2014) and Lithuania (2014–2016). In addition, the same study showed a positive correlation between the proportion of forests and the occurrence of ASF in Estonia (2015), Latvia (2015–2016), and Lithuania (2014–2016). However, in Poland and Estonia, (2014–2016), no associations were observed between the tested variables and the occurrence of ASF outbreaks [19].

There is also a hypothesis of ASF virus transmission through vectors, formulated a few years ago, but demonstrated only for soft ticks of the genus *Ornithodoros* [20]. However, the presence of these arthropods in Africa and the Mediterranean basin and the wide spread of ASF cases in Europe and Asia have led to the consideration of other possible vectors. The vector role of mosquitoes in ASFV transmission was evaluated in a study conducted in China, where no arthropods out of a total of 463 mosquitoes harvested from ASF virus-positive farms were identified as positive following the RT-PCR assessment [21]. Another study performed in Romanian pig farms (both backyard and industrial) showed the possible involvement of arthropod vectors such as *Stomoxys* spp. and *Culicoides* spp. in the transmission of the ASFV [22].

Although no association between mean altitude and ASF was identified in this study, Martinez-Lopez et al. observed an association between mean altitude and ASF in Sardinia between 1993–2009. The increases in altitude were also associated with the increase in predictability even if the number of pigs in these regions was lower than average [23]. In previous studies, the occurrence of outbreaks in domestic pigs was correlated with the distribution of cases in wild boars, indicating the link between the risk of introducing ASF virus in domestic pigs and the level of contamination of the external environment [8]. Future studies are needed to assess the relationship between the ASF outbreaks in both domestic pigs and wild boars in Romania.

Although this study provides additional data to the current knowledge, the authors acknowledge its limitations. All analyzed data provide information regarding the climatic and environmental factors influencing the occurrence of ASF outbreaks over a short period of time, large scale studies being necessary for a better understanding. Consequently, data was statistically analyzed, while no farm visits, nor other analyses, were performed. Furthermore, no exact data regarding the total number of authorized farms in each locality were available.

## 5. Conclusions

This is the first study evaluating the influence of climatic and environmental factors in ASF outbreaks in Romania. The present article concludes that most ASFV infections occurred in backyard farms and rarely in the industrial ones. The present data support the positive correlation between the number of ASFV-infected pigs and ASF outbreaks with: altitudes, average annual humidity, average seasonal temperatures (spring, summer, autumn and winter), distances from forests, lakes and rivers, geographical location, and month and season of outbreak confirmation. More data are necessary in order to have a better prediction of possible new ASF outbreaks.

## Figures and Tables

**Figure 1 animals-12-00781-f001:**
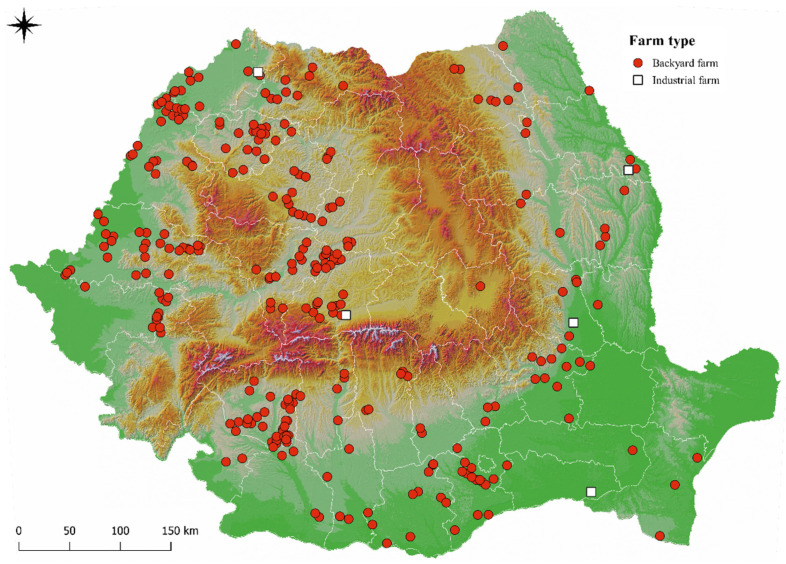
ASF outbreaks confirmed in Romania between 6 February 2020 and 2 March 2021.

**Figure 2 animals-12-00781-f002:**
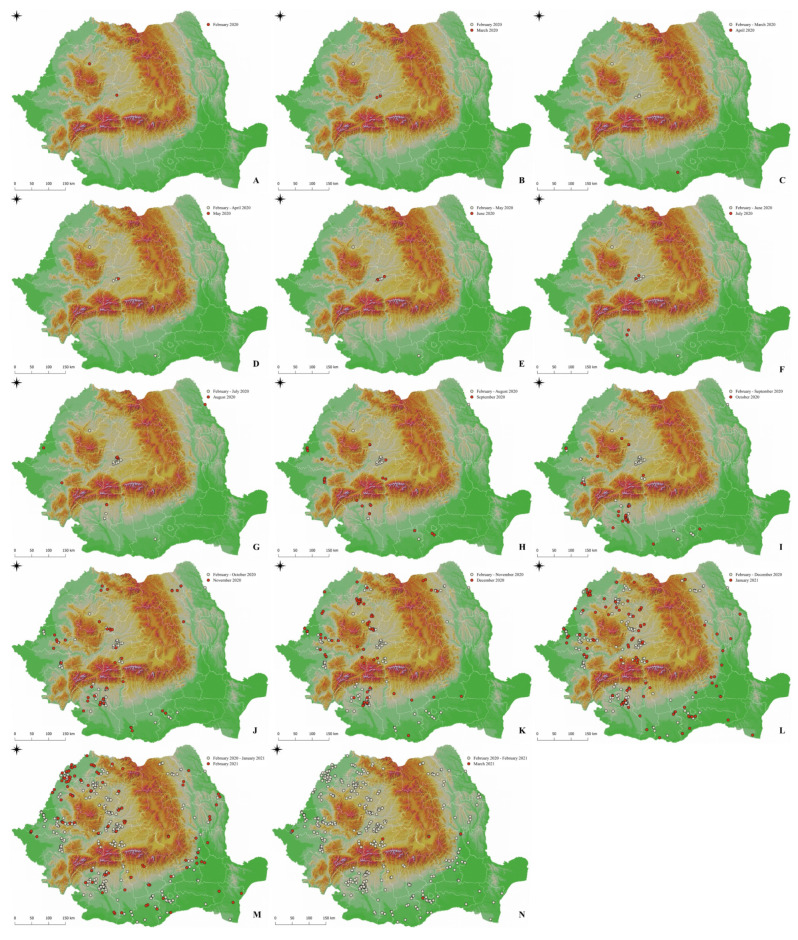
Monthly distribution of ASF confirmed outbreaks in Romania: 6–29 February 2020 (**A**), March 2020 (**B**), April 2020 (**C**), May 2020 (**D**), June 2020 (**E**), July 2020 (**F**), August 2020 (**G**), September 2020 (**H**), October 2020 (**I**), November 2020 (**J**), December 2020 (**K**), January 2021 (**L**), February 2021 (**M**), 1–2 March 2021 (**N**).

**Figure 3 animals-12-00781-f003:**
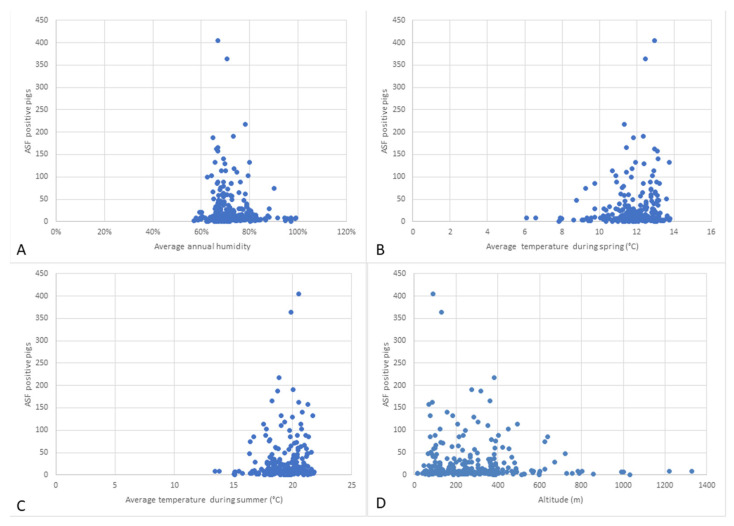
Statistical analysis for the association between the number of ASFV-positive pigs and the average annual humidity (**A**), average temperature during spring (**B**), average temperature during summer (**C**) and altitude (**D**).

**Figure 4 animals-12-00781-f004:**
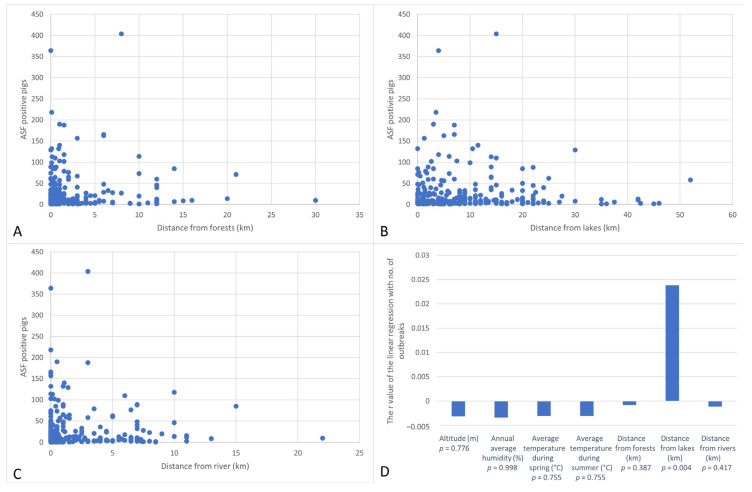
Statistical analysis for the association between the number of ASFV-positive pigs with the distance from forests (**A**), lakes (**B**), and rivers (**C**). The linear regression analysis between climatic factors (average annual humidity, temperature during spring and summer), environmental factors (altitude, the distance between a forest, lake, and river) and the number of outbreaks in each locality (**D**).

**Table 1 animals-12-00781-t001:** Distribution of ASFV-infected pigs and ASF-positive outbreaks in Romania.

County	No. of Infected Pigs in Backyard Farms	No. of Outbreaks in Backyard Farms	No. of Infected Pigs in Industrial Farms	No. of Outbreaks in Industrial Farms
Alba	448	23		
Arad	1867	43		
Argeș	46	5		
Bacău	3	2		
Bihor	630	28		
Bistrița-Năsăud	98	4		
Botoșani	25	2		
Buzău	158	12		
Caraș-Severin	4	1		
Călărași	0	0	19,908	1
Cluj	505	27		
Constanța	38	3		
Covasna	48	3		
Dâmbovița	66	4		
Dolj	132	4		
Galați	2	1		
Giurgiu	451	8		
Gorj	900	66		
Ialomița	4	1		
Iași	46	4	804	1
Ilfov	103	6		
Maramureș	287	12	2591	1
Mehedinți	210	4		
Mureș	215	13		
Neamț	33	1		
Olt	151	4		
Prahova	179	3		
Satu Mare	71	14		
Sălaj	203	29		
Sibiu	371	22	29,322	1
Suceava	86	10		
Teleorman	127	14		
Timiș	186	17		
Vaslui	23	4		
Vâlcea	20	4		
Vrancea	28	6	33,808	1

## Data Availability

The datasets supporting the conclusions of this article are included within the article and its additional files.

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
