# Peer review of "What Is the Real Influence of Climatic and Environmental Factors in the Outbreaks of African Swine Fever?"

_animals, 2022, doi:10.3390/ani12060781_

Round 1

Reviewer 1 Report

This study examines the pattern of ASF outbreaks in Romania and some interesting results are provided. However, the methodology is hard to follow and there seem to be many assumptions made about the observed epidemiology. The results are not fully validated as currently presented. There are comments about wild boar and the impact of environmental factors on wild boar habitat preference but no data is provided on wild board interactions with the backyard pigs. Is the number of pigs infected with ASF directly linked to the number of outbreaks as these two outcomes seem to be mentioned in various sentences but not in a uniform way. Was farm biosecurity and movement of animals to/from farms examined ? Perhaps a network analysis has already been done and could be mentioned in the introduction. In the discussion there is a lot of mention of the role of invertebrates in the spread of ASF but other than the soft ticks this may be a bit too speculative.

Author Response

Re: The authors thank Reviewer 1 for the revision of the submitted manuscript. A point-by-point response is given below for each comment/suggestion:

This study examines the pattern of ASF outbreaks in Romania and some interesting results are provided. However, the methodology is hard to follow and there seem to be many assumptions made about the observed epidemiology. The results are not fully validated as currently presented. There are comments about wild boar and the impact of environmental factors on wild boar habitat preference but no data is provided on wild board interactions with the backyard pigs.

Re: The authors have made some changes in the Materials and Methods section for a better understanding of the methodology. Also, new statistical data were added based on new information added in the original data base which was now included as Supplementary material. The statistical analysis was improved by the use of more tests. All new information is included in the revised submitted manuscript. Lines 267-272 were added to support the interactions between wild boars and domestic pigs.

  • Is the number of pigs infected with ASF directly linked to the number of outbreaks as these two outcomes seem to be mentioned in various sentences but not in a uniform way.

Re: Yes, the number of ASFV infected pigs and the number of ASF outbreaks are statistically linked, Lines 158-164. All phrases where this is mentioned were revised for a better understanding.

  • Was farm biosecurity and movement of animals to/from farms examined? Perhaps a network analysis has already been done and could be mentioned in the introduction.

Re: No, the farm biosecurity and movement of animals to/from farms was not examined. This article is based on data registered by the National Authorities, with no access in the outbreaks’ areas. The biosecurity level was explained based on available references and traditional behavior of the Romanian breeders.

  • In the discussion, there is a lot of mention of the role of invertebrates in the spread of ASF but other than the soft ticks this may be a bit too speculative.

Re: The authors revised the Discussion section. The mechanical role of invertebrates in the spread of ASF is currently assessed by EFSA, all data referred here are cited accordingly.

Reviewer 2 Report

This work describes an interesting research on how the Climatic and Environmental Factors could have a role in the African swine fever virus spread in Romania.

The topic is actually given the continuous expansion of the disease. The presence of ASF in a region severely damages pig production and economy, disrupts trade in pigs and porcine products and may even affect social welfare. The role of the wild boar in the transmission of the virus, the removal of infected carcasses and the Biosecurity management are focal points on the disease control. Once ASF has been introduced into a previously ASF-free area, the disease can spread among wild boar through currently recognised ASF transmission pathways. These include direct pig-to-pig interaction and indirect contact, e.g., through carcasses of wild boar that died of the disease and environmental contamination.

The current understanding of environmental ASF risk factors has implications for disease management in European in the domestic and wild population and provides further insights into ASF transmission dynamics. While climate, land cover and ASF-related factors would be difficult to control, they may provide guidance on how to allocate resources. Knowledge of human activity and wild boar-related factors might offer opportunities for direct control.

The experimental design is not complete. Preliminary data should be further investigated. The work is of some interest, but it should be expanded..In particular, some epidemiological information is also missing. In the future the conclusions that can be derived from the current knowledge about environmental ASF risk factors should be applied to disease management. I

I would suggest to implemented the tables or pictures.

In conclusion, the study presented might have a merit for publication in Animals after a minor revision.

Author Response

Re: The authors thank Reviewer 2 for the revision of the submitted manuscript. A point-by-point response is given below for each comment/suggestion:

This work describes interesting research on how the Climatic and Environmental Factors could have a role in the African swine fever virus spread in Romania.

The topic is actually given the continuous expansion of the disease. The presence of ASF in a region severely damages pig production and economy, disrupts trade in pigs and porcine products and may even affect social welfare. The role of the wild boar in the transmission of the virus, the removal of infected carcasses and the Biosecurity management are focal points on the disease control. Once ASF has been introduced into a previously ASF-free area, the disease can spread among wild boar through currently recognized ASF transmission pathways. These include direct pig-to-pig interaction and indirect contact, e.g., through carcasses of wild boar that died of the disease and environmental contamination.

The current understanding of environmental ASF risk factors has implications for disease management in European in the domestic and wild population and provides further insights into ASF transmission dynamics. While climate, land cover and ASF-related factors would be difficult to control, they may provide guidance on how to allocate resources. Knowledge of human activity and wild boar-related factors might offer opportunities for direct control.

The experimental design is not complete. Preliminary data should be further investigated. The work is of some interest, but it should be expanded..In particular, some epidemiological information is also missing. In the future the conclusions that can be derived from the current knowledge about environmental ASF risk factors should be applied to disease management.

Re: The authors included more data in the data base, now attached as Supplementary Material 1 of the article. The statistical analysis was also revised and more tests were performed in order to increase the quality of the manuscript. All new data supporting the findings are included in both Material and Methods section and Results section, respectively.

I would suggest to implemented the tables or pictures.

Re: The authors added the Supplementary Material 1, Table 1 and Figure 2 for a better understanding.

In conclusion, the study presented might have a merit for publication in Animals after a minor revision.

Re: We thank the reviewer for considering the article suitable for publication after the minor revision.

Reviewer 3 Report

The manuscript by Andrei Ungur et al. entitled " What is the Real Influence of Climatic and Environmental Factors in the Outbreaks of African Swine Fever?", it describes  outbreaks of ASF in relation with the altitude, seasonal average annual humidity, average temperature during spring and summer, distance from forests, rivers, and lakes in Romania. In general, the manuscript contributes much qualitative increment of knowledge to the state- of- the-art in this field.

General comments:

In this study, the authors compare some indexes including altitude, temperature during spring and summer. Does the number of ASF outbreaks relate with these indexes in autumn and/or winter? I am wondering whether the authors consider these factor changes to influence the ASF outbreaks.

Specific comments:

Line 117-134: I suggest that a table is added to more clearly show the data with counties, numbers etc.  

Author Response

Re: The authors thank Reviewer 3 for the revision of the submitted manuscript. A point-by-point response is given below for each comment/suggestion:

The manuscript by Andrei Ungur et al. entitled " What is the Real Influence of Climatic and Environmental Factors in the Outbreaks of African Swine Fever?", it describes outbreaks of ASF in relation with the altitude, seasonal average annual humidity, average temperature during spring and summer, distance from forests, rivers, and lakes in Romania. In general, the manuscript contributes much qualitative increment of knowledge to the state- of- the-art in this field.

General comments

In this study, the authors compare some indexes including altitude, temperature during spring and summer. Does the number of ASF outbreaks relate with these indexes in autumn and/or winter? I am wondering whether the authors consider these factor changes to influence the ASF outbreaks.

Re: The authors added the missing data of autumn and winter, which were further statistically analyzed. All new data supporting the findings are included in both Material and Methods section and Results section, respectively.

Specific comments:

Line 117-134: I suggest that a table is added to more clearly show the data with counties, numbers etc.

Re: Re: The authors added the Supplementary Material 1, Table 1 and Figure 2 for a better understanding.

Round 2

Reviewer 1 Report

The manuscript is much improved, The discussion should include some consideration of the importance of biosecurity i.e. there is often a higher level of biosecurity in commercial pig units whereas in small back yard units there are more opportunities for interaction with wild boar. There should also be more discussion of the limitations of the approach used i.e. there are likely to be a number of confounding factors which might mean that the variables examined are correlated with where pig farms are located vs disease risk per se.

Author Response

Re: The authors thank Reviewer 1 for the revision of the submitted manuscript. A point-by-point response is given below for each comment/suggestion:

  • The manuscript is much improved, The discussion should include some consideration of the importance of biosecuritye. there is often a higher level of biosecurity in commercial pig units whereas in small back yard units there are more opportunities for interaction with wild boar.

Re: The authors added lines 244-249 in the Discussion section, in which they included a comparison of the biosecurity measures between industrial and backyard farms.

  • There should also be more discussion of the limitations of the approach used i.e. there are likely to be a number of confounding factors which might mean that the variables examined are correlated with where pig farms are located vs disease risk per se

Re: The authors revised the Discussion section, mentioning the limitation of the approach in our study (lines 307-311).